# Redirected Attention and Impaired Recognition Memory during Exhaustive Cycling Has Implications for Information Processing Models of Effort-Regulation

**DOI:** 10.3390/ijerph20105905

**Published:** 2023-05-22

**Authors:** Dominic Micklewright, Bernard X. W. Liew, Steffan Kennett

**Affiliations:** 1School of Sport, Rehabilitation, & Exercise Sciences, University of Essex, Colchester CO4 3SQ, UK; bl19622@essex.ac.uk; 2Department of Psychology, University of Essex, Colchester CO4 3SQ, UK

**Keywords:** physical activity, cycling, performance, perceived exertion, cognition, recognition memory

## Abstract

Perception of internal and external cues is an important determinant of pacing behaviour, but little is known about the capacity to attend to such cues as exercise intensity increases. This study investigated whether changes in attentional focus and recognition memory correspond with selected psychophysiological and physiological parameters during exhaustive cycling. Methods: Twenty male participants performed two laboratory ramped cycling tests beginning at 50 W and increasing by 0.25 W/s until volitional exhaustion. Ratings of perceived exertion, heart rate and respiratory gas exchange measures were recorded during the first test. During the second test, participants listened to a list of spoken words presented through headphones at a rate of one word every 4 s. Afterwards, their recognition memory for the word pool was measured. Results: Recognition memory performance was found to have strong negative correlations with perceived exertion (*p* < 0.0001), percentage of peak power output (*p* < 0.0001), percentage of heart rate reserve (*p* < 0.0001), and percentage of peak oxygen uptake (*p* < 0.0001). Conclusions: The results show that, as the physiological and psychophysiological stress of cycling intensified, recognition memory performance deteriorated. This might be due to impairment of memory encoding of the spoken words as they were presented, or because of a diversion of attention away from the headphones, perhaps towards internal physiological sensations as interoceptive sources of attentional load increase with exercise intensity. Information processing models of pacing and performance need to recognise that an athlete’s capacity to attend to and process external information is not constant, but changes with exercise intensity.

## 1. Introduction

Recent models have emphasised the role of perceived exertion in effort regulation and performance [1,2,3,4]. Perceived exertion-derived modifications in pace are thought to arise from complex interactions between conscious internal physiological sensations [5,6] and external exteroceptive environmental information [7,8,9]. An assumption of these models is that, in making pacing decisions, athletes have unlimited capacity to attend to both internal sensations and external information, but this may not be the case [10,11,12].

According to the effort-related model, during conditions of low physical or psychological effort, attention can be voluntarily and easily shifted between intrinsic and extrinsic thoughts, as well as change between narrow and wide focus [13]. The model also predicts that, as the physical or psychological effort demands of an object or task increase, an involuntary shift in cognitive attention towards that object or task occurs. This is consistent with work in which greater associative thoughts occurred during high-intensity running [14]. During high-intensity physical activity, this means that attention might not be voluntarily controlled but instead is compelled to focus narrowly on associated internal physical sensations [15]. Thus, it is the effort-related attention shift from dissociative to associative thought that is the primary feature of this model. Associative thoughts, with respect to exercise, have been described as those having an inward focus on bodily sensations in contrast to dissociative thoughts having an outward focus on stimuli in the external world [16,17].

The effort-related model is supported by research in which exercise is used as the physical stressor [15,18,19,20,21,22]. In two separate studies, associative thoughts were found to be prevalent during high-intensity running [18,20]. Elite runners have also reported using associative psychological strategies during competition that is also very high-intensity [13]. In two related studies, participants were asked to verbalize their thoughts during a handgrip task to failure, as well as a cycling task performed at three different intensities (50%, 75% and 90% of VO_2MAX_). In all instances, a shift from dissociative to associative thought was found [16]. A similar shift in attention has also been found at three different intensities of rowing performed at 30%, 50% and 70% of peak power output [15].

The effort-related model [13] provides a very useful explanation of why volitional control and focus of thought changes during circumstances of physical or psychological stress, but there are opportunities to strengthen its empirical basis. There is a need to establish the continuous nature of changes in associative and dissociative thoughts because, as highlighted above, they have only really been evidenced at discrete intervals [15]. The second opportunity involves the use of rating scales, which in many of these studies were presented to the participants by an experimenter to measure perceived exertion or focus of attention. The consequence of using such scales, sometimes administered as frequently as every minute [15], is that they interfere with the continuous stream of thought that would have been uninterrupted had the scale not been presented. Put another way, presenting participants with a rating scale explicitly directs their attention towards internal states that otherwise may or may not have been attended to and thus introduces an attention-response bias.

Perceived exertion scales, such as the Borg scale [23], require respondents to reflect upon their internal sensations during physical effort and thus constitute a deliberate redirection towards associative thought. This rating–attention interaction is unlikely to be a problem during high-intensity exercise because, assuming the effort-related model is correct, associative thoughts would be prevalent anyway. However, during low and moderate levels of physical stress, the administration of a rating scale could confound natural attention processes. The importance of passively measuring perceived exertion was recognised in studies where the electromyography of frowning muscle activity during exercise was found to correspond with RPE [24,25].

The purpose of this study was to measure continuous rather than discrete changes in attention with increasing physical stress using passive measurement methods. We predicted that the ability to attend to and remember external stimuli would concurrently diminish with increases in exercise intensity and measures of internal physiological stress.

## 2. Materials and Methods

### 2.1. Participants

Twenty male participants were recruited whose mean (SD) age, stature and body mass was 21.0 (0.9) years, 178.5 (6.4) cm and 72.9 (6.9) kg. All participants were healthy and physically active: body mass index 22.9 (1.9) kg/m^2^; resting heart-rate 56.6 (5.1) b/min; and peak oxygen uptake 44.1 (9.9) mL/kg/min. Competitive or habitual recreational cyclists were excluded from the study.

### 2.2. Design

Participants performed two identical laboratory cycling tests designed to impose gradually increasing levels of physiological stress. During the first test, selected physiological and psychophysiological measures were recorded that were later correlated with recognition memory performance measured during the second cycling test. Both cycling tests were performed at the same time of day (±1 h) with a recovery interval between tests of 3–7 days. Participants were asked to refrain from eating or consuming caffeine for two hours before each test, and were asked not to perform any other exercise until they had completed both laboratory tests.

### 2.3. Cycling Ergometry Procedures

Each participant attended the laboratory twice, and, on each occasion, performed a ramped cycling test until volitional exhaustion on a Lode Excalibur electromagnetically braked cycle ergometer (Excalibur Sport, Lode B.V., Groningen, The Netherlands). Each cycling test began with a resistance of 50 W that was increased by 0.25 W/s until the participant could no longer continue. Participants were asked to maintain a pedalling cadence of 60 rpm and were instructed to remain seated on the saddle for the duration of the test. Saddle height was adjusted so that, with the crank position at bottom dead centre and the foot secured to the pedal with toe clips, the knee joint was almost in full extension (approximately 175–180°) and the sole of the foot was parallel to the ground.

### 2.4. Physiological and Psychophysiological Measures

Physiological and psychophysiological measures were only recorded during the first cycling test. This was necessary to minimise the interference to the memory recognition task, described below, that could have been caused by having to wear quite a lot of monitoring equipment, such as a facemask. Heart rate (HR), used as an indicator of cardiovascular stress, was continuously recorded using a Polar s610 heart rate monitor via a wireless chest strap (Polar Electro, Kempele, Finland), and peak HR was measured during the first cycling test. Heart rate values were transformed into percentage of heart rate reserve (HRR) using the Karvonen method [26]. 

Respiratory gas exchange measures, used as indicators of metabolic stress, were measured breath-by-breath using an online gas analyser (Oxycon Pro, Jaeger, Höchberg, Germany). Rate of oxygen uptake (VO_2_) and carbon dioxide elimination (VCO_2_) are expressed as percentages of peak oxygen uptake (VO_2PEAK_) and peak carbon dioxide elimination (VCO_2PEAK_), respectively. Respiratory exchange ratio (RER), calculated as VCO_2_/VO_2_, was also recorded during each cycling test because it is a reliable indicator of metabolic stress.

At 80 s intervals during the first cycling test, participants were asked to state their perceived exertion using the 6–20 fifteen-point RPE scale [23]. Each participant was familiarised with the RPE scale, which was administered in accordance with published standardised instructions [27]. It was necessary to only measure RPE during the first cycling test because asking participants to give ratings would distract them from the memory recognition task.

### 2.5. Recognition Memory Task

During the second laboratory attendance, approximately 3–7 days after the first test, participants repeated the cycling test described in Section 2.3 above using the same protocol and cycling ergometer settings. A recognition memory task was performed during the second cycling test but, to avoid the rating–attention interaction (described in Section 1), no physiological or psychophysiological measures were taken.

The recognition memory task involved participants listening to a verbal recording of the Toronto word pool [28] through headphones while they performed their cycling task. One word was presented every 4 s, which corresponded to a 1 W increase in cycling ergometer flywheel resistance. Participants were instructed to listen to the words through the headphones but, as with the first test, to cycle for as long as possible until they could no longer continue. Participants were allowed to rest for 15 min after they had stopped cycling and then, without prior notice, were asked to perform a recognition memory test of the Toronto word pool. The recognition test involved participants listening to the previously presented words mixed with the same number of new words that they had not previously heard. The words in the recall test were presented over headphones in a random order and at a rate of one word every 4 s; the same rate as the initial presentation of the words during the cycling test. Participants were asked to indicate on a printed list of the words whether they had heard them during the cycling test or whether they were new items. Recognition memory error was calculated as the percentage of all items presented that were correctly identified. The percentage of new items that each participant believed were presented was also calculated.

### 2.6. Data Analysis

For each individual participant, heart rate reserves and gas exchange data (VO_2_, VCO_2_ and RER) were taken from the first cycling test. For each of these measures, an average was calculated for each RPE level. Thus, if a given participant remained at an RPE score of 13 for a period of, say, 35 s, the above physiological measures were averaged across that period of 35 s. Recognition memory error during the second cycling test was calculated for the power output range that corresponded to each RPE point recorded during the first cycling test. Bland–Altman plots were used to check the similarity in performance between the two cycling tests [29].

The resulting transformed data set meant that, for each individual participant, there were multiple RPE segmented memory recognition scores and multiple corresponding data points for power and various cardiorespiratory measures. This meant that the assumption of independence, normally required for Pearson Product Moment Correlations, was violated, and therefore we analysed our data using repeated measures correlations [30] using the established RMCORR method in *R* [31]. Lower and upper 95% confidence limits were calculated using the bootstrap technique with 1000 iterations. Repeated measures correlation plots were also produced in R to show the overall regression line and colour-differentiated individual correlations.

## 3. Results

### 3.1. Cycling Test Performance

A Bland–Altman plot (Figure 1A) indicated all but one of the participants performed the same in both ramped cycling tests. In accordance with the Bland and Altman procedure [29], the data for this participant was removed from all further analyses. A revised Bland–Altman plot (Figure 1B) indicated a high level of repeatability in cycling test performance for the remaining 19 participants, which was corroborated by the high Pearson’s Product Moment Correlation coefficient value, r(18) = 0.998, *p* < 0.0001, and the low regression equation intercept value, Test 2 Power = (0.9963 × Test 1 Power) − 5.374. 

### 3.2. Test 1: RPE Correlations with Power Output and Physiological Measures

Strong positive repeated measures correlations were found between RPE and percentage of peak power output (r_rm_ = −0.984, 95% CI [−0.980, −0.988], *p* < 0.0001), percentage of heart rate reserve (r_rm_ = −0.970, 95% CI [−0.963, −0.978], *p* < 0.0001), percentage of VO_2PEAK_ (r_rm_ = −0.972, 95% CI [−0.966, −0.978], *p* < 0.0001), percentage of VCO_2PEAK_ (r_rm_ = −0.969, 95% CI [−0.962, −0.975], *p* < 0.0001), ventilation rate (r_rm_ = −0.720, 95% CI [−0.668, −0.766], *p* < 0.001), and respiratory exchange ratio (r_rm_ = −0.717, 95% CI [−0.622, −0.815], *p* < 0.001). These strong results provide the confidence needed to go on and measure correlations between test 1 physiological outcomes with test 2 memory recognition performance as determined from the power range equivalent to test 1 RPE increments. 

### 3.3. Test 2: Recognition Memory Performance Associations with Test 1 Power Output-Matched RPE and Physiological Measurements

The average recognition memory error (percentage of presented items not recognised) for the whole cycling test was 65.4 (12.1)%. The average amount of previously unheard items incorrectly identified as having been presented during the cycling test was 27 (17.5)%.

Significant repeated measures correlations were found between recognition memory performance and all physiological and psychological variables. Negative correlations were found between recognition memory error and RPE (r_rm_ = −0.713, 95% CI [−0.784, −0.636], *p* < 0.001), percentage of peak power output (r_rm_ = −0.710, 95% CI [−0.770, −0.628], *p* < 0.001), percentage of heart rate reserve (r_rm_ = −0.687, 95% CI [−0.756, −0.604], *p* < 0.001), percentage of VO_2PEAK_ (r_rm_ = −0.708, 95% CI [−0.770, −0.637], *p* < 0.001), percentage of VCO_2PEAK_ (r_rm_ = −0.707, 95% CI [−0.769, −0.625], *p* < 0.001), ventilation rate (r_rm_ = −0.493, 95% CI [−0.563, −0.420], *p* < 0.001), and RER (r_rm_ = −0.499, 95% CI [−0.611, −0.378], *p* < 0.001).

The repeated measures correlation plot for recognition memory with RPR is presented in Figure 2, for percentage of peak power output in Figure 3a, and for all ventilation and gas exchange variables in Figure 3b–f. Note that interparticipant variability causes these scatterplots to depict apparently weak associations between the variables. However, across participants, the consistency of the within-participant relationships between pairs of variables is reflected in the highly significant results of the above statistical analyses.

## 4. Discussion

Strong negative associations were found between exercise-induced physiological changes and recognition memory. As the physiological and psychophysiological stress of cycling intensified, recognition memory performance deteriorated. Consistent with the effort-related model [13], this might have been due to an involuntary attentional shift with increased cycling intensity away from the Toronto word pool listening task towards associative internal physiological sensations. An alternative explanation, given the type of recognition memory task used in our study, is that memory encoding of the spoken words was affected by increasing exercise intensity.

### 4.1. An Inverse Relationship between Recognition Memory and Exercise Intensity

Most previous work on attentional processes during physical activity concerned itself with effects on skilled tasks [32], expert performance [33], and automatic control [34]. Such studies have tended to show that attention focused on the mechanics of a task has a facilitative performance effect for novices but a debilitative effect for experts, an effect that is exacerbated by high-pressure situations [35]. Consistent with our findings, others have shown that shifts from dissociative to associative attention are critically important to performance of gross motor tasks such as running or cycling [13,15,36]. The relationship that we observed between recognition memory performance and physiological stress can be accounted for in several ways.

### 4.2. Physiological Stress and Attention: Selection, Capacity or Direction?

Classic theories of attention suggest that information is filtered out either early immediately after sensory registration [37], or later after word recognition has occurred [38]. If either of these explanations were true, we would perhaps expect to observe a threshold in the cycling intensity spectrum where recognition memory performance suddenly deteriorates due to the sudden filtering out process. However, our observations, as illustrated in Figure 2 and Figure 3a–e, show a gradual rather than sudden diminishment in recognition memory, and therefore, early filtering explanations seem unlikely. Our findings are more characteristic of attenuated filtering [39], whereby the amount of information attended to and encoded to memory gradually changes as other competing demands on attention resources emerge. The notion of competing attentional demands is accounted for in load theory of selective attention and cognitive control [40], which states that the level of perceptual load in a task determines whether any additional stimuli are perceived [41,42]. Similar perceptual loading might be achieved through the interoceptive senses, such as those generated during increasingly intense cycling, since attentional processes across the senses are highly linked [43] and possibly controlled by a single supramodal system [44,45,46,47]. In our study, it is likely that cycling-related increases in interoceptive load has led to fewer resources being available to attend to the spoken words. Furthermore, auditory tasks are performed less well if tactile attention is directed away from the source of the auditory information [48]. Therefore, it may be that increasing exercise intensity draws attention away from auditory channels towards internal physiological sensations.

### 4.3. Physiological Stress and Recognition Memory

The direction and/or the capacity of attention seems like one viable explanation for our finding of exercise-related costs to recognition memory. However, an alternative, perhaps complimentary, view is that memory processes themselves were directly affected by exercise intensity. Although our measurements did not distinguish between attention and memory, it seems quite likely that our results might have been due to an interaction between the two. This interpretation is also consistent with the suggestion of a processing gradient of memory encoding that becomes less intense as the focus of attention moves further away from the memory target [49]. Interestingly, it is thought that working memory is diminished when distractions interfere with the ability to control attention [50,51,52]. It is quite conceivable that the physiological disruptions that were observed in our study became gradually more distracting with cycling intensity that interfered with the memory of Toronto Word Pool items.

In our experiment we measured long-term recognition memory performance: A relatively large number of word items were played to participants and fifteen minutes elapsed before we presented the recognition test. However, because the circumstances of the recognition test were constant for all participants, giving them all ample recovery time, we suggest that our results are more likely to be due to disruptions to the initial perception and/or encoding phase, rather than recognition process. When objects are studied, such as the Toronto Word Pool items presented in our study, incomplete and error-prone copies of a lexical or semantic trace are stored [53]. The findings of our study perhaps indicate that the completeness of such word traces is inversely associated with cycling intensity and physiological stress. According to global-matching theories [54,55] less complete word traces have low familiarity values, meaning that they are less likely to be matched during any subsequent recognition trial. Other accounts of recognition memory exist and, as such, the details underlying cognitive mechanisms remain controversial [53,56,57,58]. Nevertheless, a degraded input will always lead to a subsequent recognition memory cost.

### 4.4. Limitations

Investigating memory recognition during exercise presented a number of methodological challenges. Ideally, the memory recognition measurements using the Toronto Word Pool auditory protocol would have been carried out concurrently with perceived exertion and various cardiorespiratory measures during the same graded cycling test. However, to do so would mean that both perceived exertion methods and cardiorespiratory measurement methods would have interfered with the task of listening to the Toronto Word Pool as presented to the participant through headphones. For example, perceived exertion is measured by asking participants to look at and give a rating, which in our study we needed to do frequently, at 80 s intervals, which corresponded to each power increase of 15 W. The required high frequency of RPE measurements would therefore have diverted participants attention away from the listening task, causing significant interference. Cardiorespiratory measurements are relatively intrusive and require participants to wear a full face mask, something which most participants will rarely, if ever, have encountered before. Breathing through a face mask and wearing headphones can cause participants to become more aware of their breathing, especially at lower-intensity exercise, thus also having the potential to interfere with the auditory listening task.

For these reasons, a limitation and constraint of this study is that it was necessary to conduct the Toronto Word Pool listening task during a separate graded cycling test to the perceived exertion and cardiorespiratory measures. Because of this, it was important to verify a high level of test–retest reliability in cycling performance between the two tests. We did this using Bland–Altman analysis, which did in fact cause us to exclude one participant from the original sample because cycling performance was not sufficiently similar.

A further limitation of our study is that it was not possible to definitively resolve the ambiguity of whether the observed intensity-related diminishment in memory recognition was due to involuntary shifts in attention or deficits in memory encoding. Changes in selective attention are conventionally detected using verbal shadowing techniques in which participants are asked to verbalise words that are presented to them, for example, either audibly through headphones or visually on a screen. It is inferred that when a participant is unable to verbalise, this is a symptom of deficits in attentional capacity or direction. However, it was not possible to use this technique in our study because verbalisation interferes with breathing during cycling, and at higher intensities verbalisation itself becomes very difficult. Thus, further ambiguity would exist, because any changes in verbalisation could either be due to intensity-related breathing effects or attentional effects. Developing an attention-monitoring protocol that is suitable for use during high-intensity exercise is methodologically challenging, but is something that will need careful consideration in future studies.

## 5. Conclusions

The results show that, as the physiological and psychophysiological stress of cycling intensified, recognition memory performance deteriorated. This might be due to impairment of memory encoding of the spoken words as they were presented, or because of a diversion of attention away from the headphones, perhaps towards internal physiological sensations, as interoceptive sources of attentional load increase with exercise intensity.

The capacity of an athlete to attend to various types of internal and external information has not really been considered in most information-processing models of pacing a performance. While the direct effect of attention shifts of pacing and performance were not measured in this study, the results nevertheless show that, when they are very exerted, athletes are more likely to miss external cues. Information processing models of pacing and performance need to recognise that an athlete’s capacity to attend to and process external information is not constant but changes with exercise intensity and, as previously predicted, may not be under volitional control [11,13]. In view of our findings, more explicit examination of attention capacity and effort regulation is needed.

## Figures and Tables

**Figure 1 ijerph-20-05905-f001:**
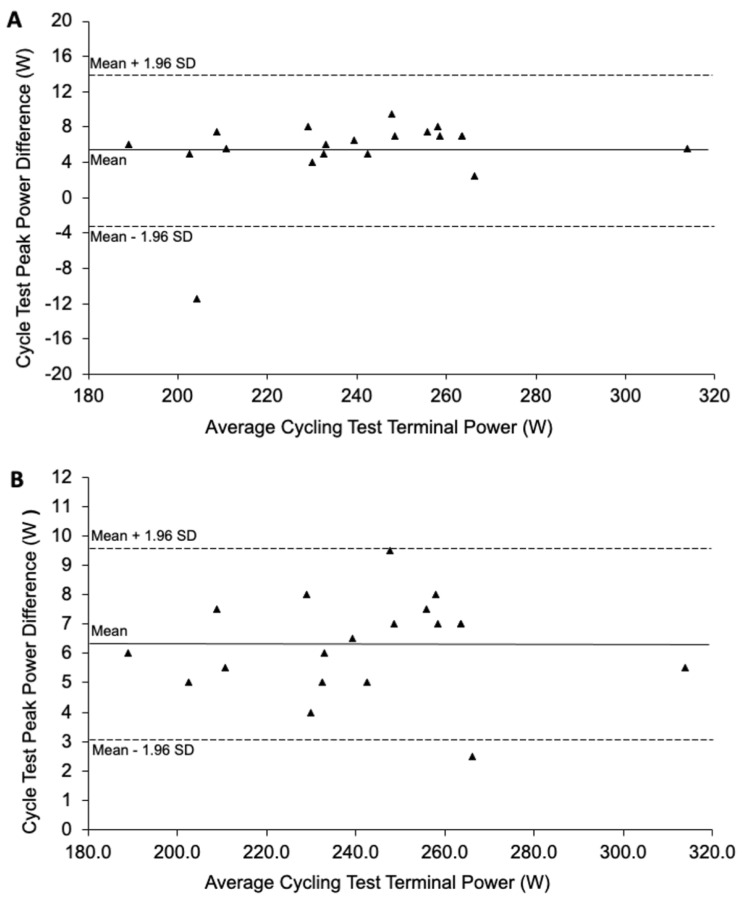
Bland-Altman plot indicating all but one participant (indicated with triangles) had similar cycling test performance in both trails (**A**) and a revised plot with the participant with difference performances removed (**B**).

**Figure 2 ijerph-20-05905-f002:**
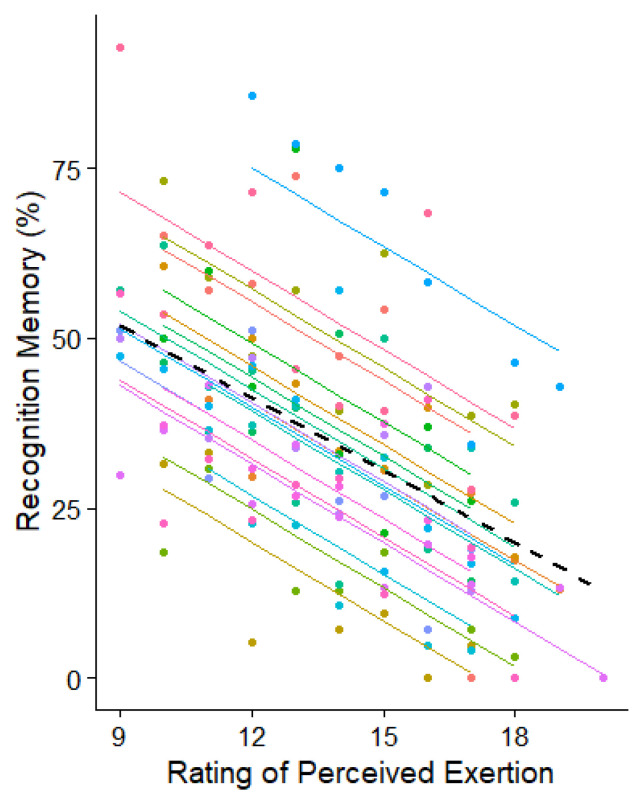
Repeated measure correlation plot between memory recognition and perceived exertion. Each participant is indicated with a different colour.

**Figure 3 ijerph-20-05905-f003:**
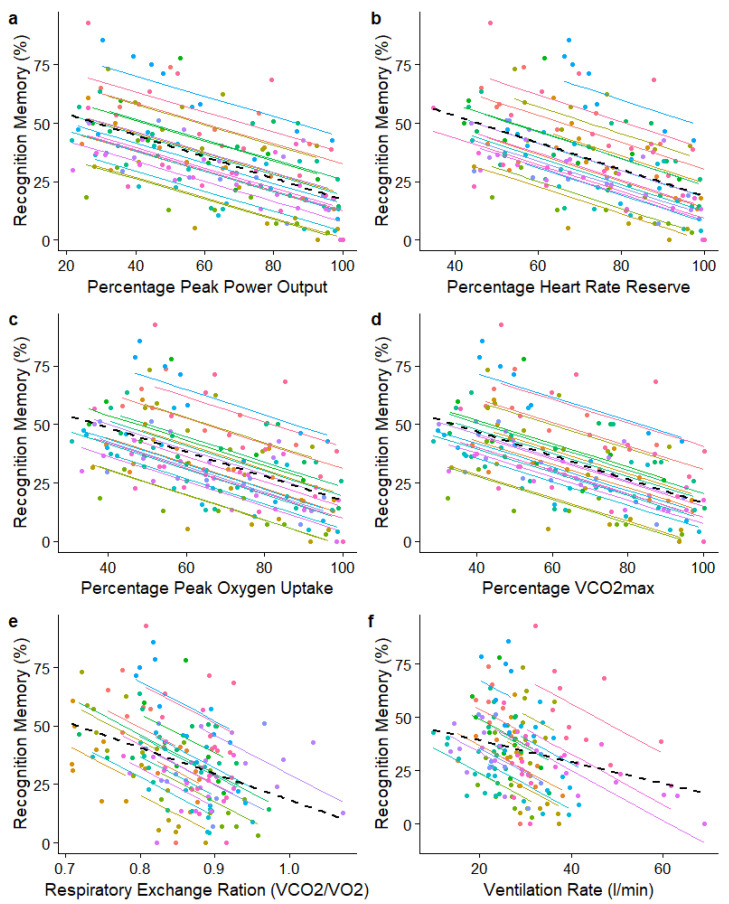
Repeated measures correlation plots between memory recognition and peak power output (**a**), relative heart rate reserve (**b**), percent of oxygen uptake (**c**), percent of peak carbon dioxide production (**d**), respiratory exchange ration (**e**), and ventilation rate (**f**). Each participant is indicated with a different colour.

## Data Availability

Data is available on request of the Corresponding Author.

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
