# Peer review of "Redirected Attention and Impaired Recognition Memory during Exhaustive Cycling Has Implications for Information Processing Models of Effort-Regulation"

_ijerph, 2023, doi:10.3390/ijerph20105905_

Round 1

Reviewer 1 Report

This is an interesting study, and the authors have developed an effective method to explore the interactions between exercise intensity and episodic memory. The authors discuss implications of the interactions between exercise and memory in terms of effort regulation, which is a novel perspective. This discussion is a useful integration of research areas. The novelty of the work here is that exercise intensify has been progressively increased during a continuous memory task.  This is in contrast to previous research that typically compares separate conditions of intensities. This has allowed for a demonstration of the progressive effects of exercise intensity on encoding processes. I commend the authors for their efforts in completing this study design.

I have two main concerns at this stage, and I provide feedback in the hope of assisting the authors further develop the work such that the findings can be presented.

Firstly, this study speaks to a large body of work exploring the associations between acute concurrent exercise and memory processes. However, this work is not referred to here. In terms of theoretical perspectives, methodological considerations, and findings, this work seems critical for inclusion in the present study.

Secondly, as the primary dependent variable, further details of the administration of the memory task are required.

I expand on these considerations below:

Existing evidence sand perspectives.

1.       Acknowledging some of the broader perspectives regarding concurrent acute exercise and cognition functioning would benefit the framing of the work.  At present this reads as narrow; primarily focused on the impact of internal physiological sensations. Consider for example, the impact of concurrent exercise intensity on core executive functions. See:

Zheng, K., Zou, L., Wei, G. and Huang, T., 2021. Concurrent Performance of Executive Function during Acute Bouts of Exercise in Adults: A Systematic Review. Brain sciences, 11(10), p.1364.

2.       The impact of the timing and intensity of acute exercise on memory seems similarly important here.  Not least the observations that when acute exercise occurs during memory encoding, memory function is often impaired – an effect exaggerated by increasing intensity (as seen here). But the evidence to-date suggests this may be a product of cognitive resources being allocated to sustain the primary movement itself. Can the authors incorporate these perspectives into their discussion? See:

Roig, M., Thomas, R., Mang, C.S., Snow, N.J., Ostadan, F., Boyd, L.A. and Lundbye-Jensen, J., 2016. Time-dependent effects of cardiovascular exercise on memory. Exercise and sport sciences reviews, 44(2), pp.81-88.

Loprinzi, P.D., Blough, J., Crawford, L., Ryu, S., Zou, L. and Li, H., 2019. The temporal effects of acute exercise on episodic memory function: Systematic review with meta-analysis. Brain sciences, 9(4), p.87.

Dietrich, A., & Audiffren, M. (2011). The Reticular-Activating Hypofrontality (RAH) Model of Acute Exercise. Neuroscience & Biobehavioral Reviews, 35(6), 1305-1325.

3.       Further consideration of theoretical perspectives within this area of work should compliment some of the attentional perspectives already offered by the authors of the present work. The discussion of dual-task demands of motor control concurrently with updating sensory feedback seems relevant here given the authors emphasis of attentional explanations. At present the authors focus on interoceptive information processing as the critical process, but no mention is made of the increasingly demanding nature of maintaining the primary motor task itself. Increasingly challenging exercise intensities results in various changes other than psychophysical perception alteration, not least impaired motor control, metabolic changes, as well as fatigue. Indeed, participants were not regular cyclists, suggesting relatively novel motor demands as well.  Recent efforts by Tomporowski and Qazi to frame the theoretical perspectives appear relevant here. Loprinzi (2019) point to the “redistribution of neuronal resources away from key memory-related brain structures” when exercise occurs during memory encoding. Can these be incorporated in the work?

Tomporowski, P.D. and Qazi, A.S., 2020. Cognitive-motor dual task interference effects on declarative memory: A theory-based review. Frontiers in psychology, 11, p.1015.

Loprinzi, P.D., 2019. An integrated model of acute exercise on memory function. Medical hypotheses126, pp.51-59.

4.       A consideration of how the high-intensity exercise experienced in the latter stages of the fitness test negatively impacted encoding of words could be acknowledged. See:

Frith, E., Sng, E. and Loprinzi, P.D., 2017. Randomized controlled trial evaluating the temporal effects of high‐intensity exercise on learning, short‐term and long‐term memory, and prospective memory. European Journal of Neuroscience, 46(10), pp.2557-2564.

5.       In the discussion, the authors do suggest that encoding of word memory traces is impacted by concurrent exercise. But can the authors supplement this discussion with existing work on source encoding and exercise? For example,

Soga, K., Kamijo, K. and Masaki, H., 2017. Aerobic exercise during encoding impairs hippocampus-dependent memory. Journal of Sport and Exercise Psychology, 39(4), pp.249-260.

Regarding the methods – further information will help the reader understand the tasks used here:

6.       Further information is needed here regarding how participants were instructed to engage with the encoding phase of the memory task. Were the participants aware they would be assessed on the words that were presented to them?  The phrase “without prior notice” (Line 141) regarding the recognition phase leaves me unsure on this. Was it the encoding – recognition time interval that was uncertain to participants?  The earlier instructions suggest that they were instructed to listen to the words whilst trying to cycle for as long as possible (Line 138) – but were participants specifically instructed to try to remember the words as best as they could to facilitate later recall, or was this presentation of words described as incidental to their exercise task?

7.       Was there any process to the selection and ordering of the presented words (and later distractor words) from the Toronto word pool? In terms of syllables, frequency, imagery, and concreteness.

8.       How many words were presented in the task across participants?

9.       Was the recognition task individually tailored to the participants word exposure. I imagine that it was. Some description of the process of this may be useful for the reader.

10.   Give the critical impact post-encoding periods have on later recall, can the authors provide information on what participants did during the 15 minutes of rest.

11.   Can the authors provide a rationale as to why only recognition memory was assessed? Could free recall have been a useful approach here?

Other Considerations

12.   Title: Redirected attention and impaired recognition memory during exhaustive cycling has implications for information processing models of effort-regulation. Do the methods and findings allow for the stating of redirected attention in the title?  Attention is not measured as a DV, whereas the main IV (intensity) is also not mentioned in the title.

Introduction

13.   Large parts of the introduction cover experiences of associative and dissociative thoughts during exercise. Whilst this is relevant, it could be framed within a discussion of the impact of acute exercise on memory encoding processes as outlined above.

Methods

14.   For unfamiliar readers, can the authors interpret the relative fitness levels of participants from the fitness test values reported. Fitness is a key moderator of the exercise-cognition relationship.

15.   Line 130: repetition of “approximately 3-7 days after the first test”

16.   Clarify DV - Recognition error/performance: this is stated conflictingly in numerous places. “Recognition memory error was calculated as the percentage of all items presented that were correctly identified” (line 148). Or “Recognition memory error (percentage of presented items not recognised)” (Line 188). The latter appears to be a measure of error, the former a measure of performance. Figure labels refer to performance. Lines 193 – 200 refer to both recognition memory performance and recognition memory error.

Discussion

17.   Paragraph: 4.2. An Inverse Relationship between Recognition Memory and Exercise Intensity. The early citations within this paragraph concern motor skill acquisition and the break-down of skills under pressure. At present, this content does not immediately seem relevant to the targeted content the subtitle suggests.  The later citations focus on attentional allocation to sensations of exertion associated with exercise intensity. These seem relevant, but could be linked to the evidence for the effects of exercise on memory.

18.   Line 249 “Although our measurements do not distinguish between attention and memory, it seems quite likely that our results might have been due to an interaction between the two.” Can the authors clarify their thoughts here. The primary DV is a measure of memory, whilst there appears to be no measure of attentional processes.

19.   Line 261: “we suggest that our results are more likely to be due to disruptions to the initial perception and/or encoding phase, rather than recognition process”. See work such as Loprinzi (2019) above for perspectives on this conclusion.

Author Response

Many thanks for your helpful comments and giving up your time to review our work. We have responded to your feedback in the attached document. Many thanks.

Reviewer 2 Report

1.      Overall, the paper’s aim is very interesting. However, I found a few issues that need to be fixed before an eventual publication. Some of those are related to editing, which could be found as minor issues, However, the rest of those are rather major.

Introduction:

2.      “In all instances, a shift from dissociative to associative  57 thought was found.16” You have described results from “two related studies”, however, provided a reference to just one. Please note also the need for fixing the editing part of proper noting a reference according to the journal’s style. In addition, if You have an idea how to explain “dissociative” and “associative”  in one sentence, please give it a try, If not, then You can omit this issue, as the Introduction is a very long one anyway.

3.      The aim “The purpose of this study was to measure continuous rather than discrete changes in  82 attention with increasing physical stress using passive measurement methods. We pre- 83 dicted that the ability to attend to and remember external stimuli would concurrently di- 84 minish with increases in exercise” seems to be too general. Please be more precise here.

Materials and methods:

4.      Competitive or habitual recreational cyclists  91 were excluded from the study” so how would You describe the sample? Maybe You can draw conclusions from their baseline Vo2peak? Do You think that their baseline VO2peak might influence observed results? If so, please add it to the limitations of the study subsection in the discussion and/or the need for further examination

5.      “For each participant, Pearson’s Product Moment  162 Correlation coefficients were calculated” please not that regression/Pearson correlation, assume independence of error between observation, for instance please see https://doi.org/10.3389/fpsyg.2017.00456. Therefore you have applied a proper method for analyzing for analysis data from Your experimental model?

6.      “resulting individual r-values were sub- 163 jected to a single-sample t-test across the participant group, which revealed whether the  164 correlations were significantly greater, or less, than zero” I am not fully following the sentence, I believe that the method applied might be valid, however the description/wording should be improved. what does exactly zero means? is that was Your hypothesis? if so, how it was tested? I believe that it is not the underlying hypothesis of any t test. In addition, have You checked the assumption for the t-test? What was the “participant group” exactly here?

7.      “Mean r-values (rmean) and mean  165 r2-values (r2mean)” how did you calculate r squared, please describe it in a few words

Results:

8.      “Test 2 Power=(0.9963xTest 1 Power)–5.374” how the test power was calculated? Could You describe the method in m & m section?

9.      “Strong positive correlations were found between RPE against all of the following

10.   measures: percentage of peak power output (rmean=.992, r2mean=.984, t(18)=753, p<.0001 )” where the result and following results come from? What groups are compared here?

11.   “ These strong results provide the confidence needed to go on and measure cor-relations between test 1 physiological outcomes with test 2 memory recognition perfor-mance as determined from the power range equivalent to test 1 RPE increments.” I am lost here. What does it mean that results are "strong" ? even if so, why did You chose particular steps in further analysis based on results presented so far?

12.   “The average recognition memory error (percentage of presented items not recog-

13.   nised) for the whole cycling test was 65.4(12.1)%. “ Is that mean and SD provided here?

14.   “Note that interpar- 202 ticipant variability causes these scatterplots to depict apparently weak associations be- 203 tween the variables.  “ Because You have not denoted if data points are coming from the same participants, it is not visible on the graph. Maybe You can do some color coding with each color denoted to the same participant in different time-points?

15.   in the highly significant results  “ what highly means? Please omit using such language if possible

Discussion:

16.   “As the physiological and psychphysiological stress of cycling inten- 210 sified, recognition memory performance deteriorate ed. “Please note a typo in “psychphysiological” Adding line of best fit would be helpful to confirm this conclusion. Are You sure that the relationship here is linear? Is that was Your hypothesis regarding all of the measured variables? D You think that increasing RER n particular participant from 0.7 to 0.8 or to 0.9 would affect cognitive function/attention to the same extent as increasing RER from 1.2 to 1.3? or form 1.3.to 1.4?

17.   “Classic theories of attention suggest that information is filtered out either early im- 227 mediately after sensory registration [35], or later after word recognition has occurred [36].  228 Since we observed a gradual rather than sudden diminishment in recognition memory  229 (Figure 2), both explanations seem unlikely” are You sure that this statement is true? What kind of result from Your data analysis is an evidence for this statement?

Conclusions:

18.   “This might be due to impair- 274 ment of memory encoding of the spoken words as they were presented, or because of a 275 diversion of attention away from the headphones, perhaps towards internal physiological  276 sensations, as interoceptive sources of attentional load increase with exercise intensity.  277” this part is Your speculation, not conclusion

19.   “The capacity of an athlete to attend to various types of internal and external infor- 278 mation has not really been considered in most information-processing models of pacing a  279 performance.” here the study limitation description starts. please move it to a separate paragraph in the discussion. please also extend this section. What else study limitations could be noted, especially looking at the study model and material and methods applied?

Author Response

(The authors gave the same response as above.)

Round 2

Reviewer 2 Report

Dear Authors,

From what I see, there are no answers on most of the poroposes suggestions in the first round of Review. For instance, why there is no separate paragraph with study limitations? Why this paragraph has not been extended, as suggested? Another thing it with the applied statistical analysis method: I do belive that the assumption of independence of observations in linear regression is not much about how data look like. It is about how the data was collected. in my humble opinion, if there are repeated measurments on the same participants it means that the assumption is violated. Please refer to the local statistician, maybe he or she would help in the data analysis process?

Author Response

Reviewer 2:

From what I see, there are no answers on most of the poroposes suggestions in the first round of Review. For instance, why there is no separate paragraph with study limitations? Why this paragraph has not been extended, as suggested?

Response: We have now added a new paragraph with study limitations as suggested.

Another thing it with the applied statistical analysis method: I do belive that the assumption of independence of observations in linear regression is not much about how data look like. It is about how the data was collected. in my humble opinion, if there are repeated measurments on the same participants it means that the assumption is violated. Please refer to the local statistician, maybe he or she would help in the data analysis process?

Response: Thank you. We have now updated the methods, results and figures using repeated measures correlations, all of which were conducted in R.